
# Evaluating downscaling methods of GRACE data: a case study over a fractured crystalline aquifer in South India

Claire Pascal[1], Sylvain Ferrant[2], Adrien Selles[2], Jean-Christophe Maréchal[2], Abhilash Paswan[3], and Olivier Merlin[1]

[1]Centre d'Étude Spatiale de la BIOsphère,(CESBIO-UPS-CNRS-IRD-CNES-INRAE), 18 av. Ed. Belin, Toulouse CEDEX 9, 31401, France
[2]Bureau de Recherches Géologiques et Minières (BRGM), Université de Montpellier, 1039 rue de Pinville, Montpellier, 34000, France
[3]National Geophysical Research Institute, CSIR, Hyderabad, India

**Correspondence:** Claire PASCAL (claire.pascal@univ-tlse3.fr)

**Abstract.** GRACE (Gravity Recovery and Climate Experiment) and its follow-on mission have provided since 2002 monthly anomalies of total water storage (TWS), which are very relevant to assess the evolution of groundwater storage (GWS) at global and regional scale. However, the use of GRACE data for groundwater irrigation management is limited by their coarse ($\simeq 300$ km) resolution. The last decade has thus seen numerous attempts to downscale GRACE data at higher – typically several

tens of km – resolution and to compare the downscaled GWS data with in situ measurements. Such comparison has been classically made in time, offering an estimate of the static performance of downscaling (classic validation). The point is that the performance of GWS downscaling methods may vary in time due to changes in the dominant hydrological processes through the seasons. To fill the gap, this study investigates the dynamic performance of GWS downscaling by developing a new metric for estimating the downscaling gain (new validation) against non-downscaled GWS. The new validation approach is tested over

a 113,000 km$^2$ fractured granitic aquifer in South India. GRACE TWS data are downscaled at 0.5° ($\simeq 50$ km) resolution with a data-driven method based on random forest. The downscaling performance is evaluated by comparing the downscaled versus in situ GWS data over a total of 38 pixels at 0.5° resolution. The spatial mean of temporal Pearson correlation coefficient (R) and root mean square error (RMSE) are 0.79 and 7.9 cm, respectively (classic validation). Confronting the downscaled results with the non-downscaling case indicates that the downscaling method allow a general improvement in terms of temporal agreement

with in situ measurements (R = 0.76 and RMSE = 8.2 cm for the non-downscaling case). However, the downscaling gain (new validation) is not static. The mean downscaling gain on R is about +30% or larger from August to March including both wet and dry (irrigated) agricultural seasons and falls to about +10% from April to July, during a transition period including the driest months (April-May) and the beginning of monsoon (June-July). The new validation approach hence offers a standardized and comprehensive framework to interpret spatially and temporally the quality and uncertainty of the downscaled GRACE-derived

GWS products, supporting the future efforts on GRACE downscaling methods in various hydrological contexts.





# 1 Introduction

Groundwater is an essential resource for irrigation, especially in arid and semi-arid areas. Aquifers have suffered depletion in several areas of the world these last decades and this resource is expected to be scarcer in the future (Wada et al., 2012). Monitoring and cautious management of this resource is therefore crucial. Groundwater monitoring is traditionally achieved
with networks of observation wells, but this can be challenging due to their sparse coverage, the punctual nature of the data, and the progressive abandonment of some wells or measurement difficulties and bias (Hora et al., 2019). In the meantime, new techniques for water storage monitoring have emerged with the Gravity Recovery and Climate Experiment (GRACE) satellite mission of US and Germany space agencies (NASA and DLR). The twin satellites of the GRACE mission were launched in 2002, and the continuity of the mission as covered by the GRACE Follow-On mission (GRACE-FO) launched in 2018.
The gravimetric data retrieved from these missions have provided spatialized monthly anomalies of total water storage (TWS) for two decades, available worldwide. GRACE data were widely used in hydrology to study long term evolution of TWS or groundwater storage (GWS) by removing the contributions of other surface and sub-surface compartments from GRACE TWS, at global and regional scales (Breña-Naranjo et al., 2014; Cao and Roy, 2020; Frappart et al., 2019; Papa et al., 2015; Rodell et al., 2018; Rzepecka and Birylo, 2020; Tiwari et al., 2009; Zhang et al., 2020). Nevertheless, their application at local
scale for agricultural purposes remains limited due to the very low native resolution (about 400 km) of GRACE observations (Schmidt et al., 2008; Tapley et al., 2004).

During the past decade or so, several studies have proposed methods to downscale GRACE TWS data to obtain GWS maps at a spatial resolution (typically several tens of km) higher than that of GRACE observations. Those downscaling approaches can be separated in two categories: model-based downscaling and data-based downscaling (also referred in the literature as
"dynamic downscaling" and "statistical downscaling" respectively). The model-based downscaling approach consists in assimilating GRACE TWS data in physically-based land surface or hydrological models to obtain GWS at the temporal and spatial resolution of the model, which is generally higher than GRACE's (Girotto et al., 2016; Houborg et al., 2012; Nie et al., 2019; Schumacher et al., 2018; Tian et al., 2017; Zaitchik et al., 2008). Yet this approach suffers from (i) the discrepancy between GRACE and model input data resolutions and (ii) the limitations inherent to models: model hypothesis and parametrization,
the uncertainty of meteorological forcing, and particularly the lack of representation of anthropogenic processes such as crop irrigation (Long et al., 2013). The data-based downscaling approach consists in (i) deriving a statistical model of TWS from ancillary data available at high resolution (HR), (ii) calibrating it at low resolution (LR), (iii) applying it at HR, and (iv) removing the contribution of surface and soil moisture water stocks to isolate GWS. This data-driven approach rests on the hypothesis that the hydrological/physical processes that link those variables are identical at all resolutions (Ali et al., 2021; Jyolsna et al.,
2021; Karunakalage et al., 2021; Sahour et al., 2020; Seyoum and Milewski, 2017; Vishwakarma et al., 2021; Zhang et al., 2021a, b). In the literature, data-driven methods have been used to downscale GRACE data at various scales, either at the watershed scale for a thematic approach as in Seyoum and Milewski (2017) (5,000 km$^2$ to 20,000 km$^2$), or grid-based. The downscaling resolution for data-based techniques is often limited by the coarsest resolution among the predictors : rainfall from the Tropical Rainfall Measuring Mission (TRMM) or model outputs from the NASA's Global Land Data Assimilation





System (GLDAS) at 0.25° (Ali et al., 2021; Jyolsna et al., 2021; Ning et al., 2014; Seyoum et al., 2019; Zhang et al., 2021a),

the NASA's North American Land Data Assimilation System (NLDAS) model at 0.125° (Sahour et al., 2020), the Ecological

Assimilation of Land and Climate Observations (EALCO) model at 5 km (Zhong et al., 2021) or evapotranspiration from the

Moderate Resolution Imaging Spectroradiometer (MODIS) at 2 km (Yin et al., 2018). Even finer resolution can be targeted

when using interpolation based methods, up to the kilometer (Zhang et al., 2021a; Zuo et al., 2021).

To evaluate the GRACE data downscaled from the above approaches, different strategies have been used. Table 1 lists the

validation methods used in recent papers downscaling GRACE with either model-based or data-based approaches. For both

method categories, the validation of downscaled GWS mostly relies on the in situ measures of groundwater levels (GWL),

converted or not into GWS anomalies using a specific yield (Sy) representative of the study area. Note that the GWS simulated

by models was occasionally used as reference (Houborg et al., 2012; Seyoum and Milewski, 2017). In most studies, the quality

of the downscaled GWS data set is evaluated by comparing its time series with that of GWL or GWS derived from in situ

measurements for each HR unit (spatialized – HR pixel – or localized – observation well) with one or several metrics including

coefficient of determination ($R^2$) or Nash-Sutcliffe efficiency coefficient (NSE), Pearson correlation coefficient (R), root mean

squared error (RMSE) and mean absolute error (MAE) (Ali et al., 2021; Jyolsna et al., 2021; Karunakalage et al., 2021; Sahour

et al., 2020; Yin et al., 2018; Zhang et al., 2021b, a; Zuo et al., 2021). In those studies, the downscaling procedure is considered

efficient if those metrics fall within an acceptable range, or if the downscaled product qualitatively restitutes the long-term

trends of in situ data. The point is that any downscaling method can improve or degrade the accuracy of GRACE data at

the targeted downscaling resolution depending on (i) the sub-pixel spatial variability of TWS/GWS and (ii) the uncertainties

in input model parameters and forcing. Therefore, quantifying the improvement against the GRACE data at their original

resolution is crucial to properly evaluate downscaling methods. Among the 14 data-based methods listed in Table 1, only a

few studies (Chen et al., 2019; Ning et al., 2014; Zhang et al., 2021b; Zhong et al., 2021) quantify the improvement of the

temporal agreement with in situ data of downscaled product over the original LR data. Regarding the model-based approaches,

all of them evaluate the temporal agreement of the downscaled GWS with in situ data against open loop outputs (without the

assimilation of GRACE data), but the results of the comparison against the LR GRACE TWS are not presented. Note that the

primary goal of the latter methods is to improve the model simulations using GRACE data, and not specifically to downscale

GRACE data, even though equivalence between both objectives may be argued.

For each downscaling method, Table 1 also indicates whether the evaluation of the downscaled data set is undertaken in

time or in both time and space. Zhong et al. (2021) is the only one proposing a validation strategy combining time and space

dimensions, by measuring the improvement of RMSE and R (with monthly in situ data) from LR to downscaled GWS using 42

observation wells within the GRACE pixel and for all months of the time series. This validation approach thus combines spatial

and temporal evaluations, but does not isolate their individual contributions. In particular, to the knowledge of the authors, none

of the previous studies have specifically evaluated the capability of downscaled products to restitute the GWS spatial variations

within the GRACE pixel at the temporal observation scale (one month in our case).

Another issue in the application and validation of current downscaling studies is the scale at which the GRACE data are

used at input. The combination of the ground tracks of GRACE twin satellite over a period of one month allows a native spatial





resolution of 300 to 400 km for GRACE data, both for spherical harmonic (Schmidt et al., 2008; Tapley et al., 2004) and mascon
solutions from the Jet Propulsion Laboratory (JPL) (Watkins et al., 2015). The GRACE TWS grids are however provided with
scaling factors with a resolution of 1° and 0.5° for harmonic (Landerer and Swenson, 2012) and JPL mascon (Wiese et al.,
2016) solutions respectively. Such scaling factors have been originally designed to restore the lost signal of GRACE due to
postprocessing, and to allow for averaging the 1° or 0.5° resolution oversampled TWS data over user-defined regions with a
minimum extent similar a 300-400 km resolution GRACE pixel (Landerer and Swenson, 2012). In particular, scaling factors are
not expected to efficiently downscale GRACE TWS data as neighbouring pixels are highly dependent (Landerer and Swenson,
2012). Yet, many studies directly use GRACE harmonics solutions at 1° resolution (Ali et al., 2021; Jyolsna et al., 2021;
Karunakalage et al., 2021; Ning et al., 2014; Seyoum et al., 2019; Yin et al., 2018; Zhang et al., 2021b, a; Zuo et al., 2021) or
mascons solution at 0.5° resolution (Karunakalage et al., 2021; Nie et al., 2019; Tian et al., 2017) as LR input data, which is
way finer than their actual resolution. There is no study evaluating the uncertainty in downscaled GRACE data associated with
the above assumption i.e. neglecting the scale discrepancy between the actual resolution of GRACE observations and the grid
size of the delivered oversampled GRACE data.

In this context, the objective of this study is to propose a consistent and complete validation framework covering the spatial
and temporal aspects to quantify the supplementary information of downscaled GWS from GRACE compared to the LR
original data. We test this framework on GRACE data downscaled over a granitic aquifer of 113,000 km$^2$ in the Telangana
State, in South India. We use a data-based approach to downscale GRACE mascon solution RL06M at a 0.5° resolution with
two different models: multilinear regression model and random forest. We also use this validation framework to evaluate the
downscaling potential of the scaling factor at 0.5° resolution provided with the mascon solution (hence evaluating the choice
of using the GRACE data oversampled at 0.5° resolution as a downscaled product). We compare the conclusions drawn from
the classic validation techniques and the new validation framework proposed in this study.

## 2 DATA AND STUDY AREA

### 2.1 Study area

The Telangana State is a highly irrigated and densely populated (about 335 inhabitants per km$^2$ in 2020 according to the Unique
Identification Authority of India (UIDAI)) region in South India, covering 114,800 km$^2$. It is dominated by a semi-arid climate,
where the monsoon precipitation occurs between July and October and ranges from 540 to 1,300 mm with a mean of 879 mm
(Indian Meteorological Department). The strong water demand in this area for domestic uses and the irrigation of two growing
seasons a year is met with the surface water stored from monsoon rainfall and groundwater. The majority (67,000 km$^2$) of the
State is a shallow fractured granitic aquifer characterized by high fluctuations due to water pumping (Maréchal et al., 2006). It
is usually composed of two layers: the first layer is saprolite, with a high effective porosity (Sy of 10%), extending up to ten
to fifteen meters, and it is followed by a layer of fractured granite with a low capacity (Sy around 1%) (Dewandel et al., 2017;
Maréchal et al., 2006). This aquifer has a low capacity but strong dynamics as it fills and empties almost completely every
year with monsoon rainfall and intense pumping. While continuous groundwater depletion has been observed with GRACE or



**Table 1.** Validation strategies of existing – either data-based (D) or model-based (M) – downscaling methods of GRACE data. The downscaling method is either data-based (D) or model-based (M). The two resolutions reported are the initial resolution of GRACE data (GRACE) and the target downscaling resolution (Target). GWS: in situ groundwater storage. GWL: in situ groundwater level. GWS: in situ-derived groundwater storage. The column Comp. indicates if error statistics of the downscaled product are compared with those of another reference product: GRACE data at original low resolution (LR) or model run in open loop (OL).

| Reference | Downscaling method | Resolution | | Validation data | | Validation metric | | | | Validation in | | Comp. |
|---|---|---|---|---|---|---|---|---|---|---|---|---|
| | | GRACE | Target | in situ | model output | R | R² | RMSE | Trend changes | Time | Space | |
| (Ali et al., 2021) | D | 1° | 0.25° | GWS | | X | X | X | | X | | |
| (Chen et al., 2019) | D | 1° | 0.25° | GWL | | X | | | | X | | LR |
| (Jyolsna et al., 2021) | D | 1° | 0.25° | GWS | | X | | X | | X | | |
| (Karunakalage et al., 2021) | D | 1° and 0.5° | 0.25° | GWL | | | | | X | X | | |
| (Ning et al., 2014) | D | 1° | 0.25° | GWL | | | X | | | X | | LR |
| (Sahour et al., 2020) | D | 13,700 to 59,200 km² | 0.125° | GWL | | X | | | | X | | |
| (Seyoum et al., 2019) | D | 1° | 0.25° | GWS | | X | X | | | X | | |
| (Seyoum and Milewski, 2017) | D | 500,000 km² | 5,000 to 20,000 km² | GWS | GWS | X | | X | X | X | | |
| (Vishwakarma et al., 2021) | D | 62,518 to 4,672,876 km² | 0.5° | | | | | | | | | |
| (Yin et al., 2018) | D | 1° | 2 km | GWL | | X | | | | X | | |
| (Zhang et al., 2021a) | D | 1° | 0.25° | GWL | | X | X | X | X | X | | LR |
| (Zhang et al., 2021b) | D | 1° and 0.25° | 1 km | GWL | | X | X | X | | X | | LR |
| (Zhong et al., 2021) | D | 3° | 5 km | GWS | | X | X | X | X | X | X | LR |
| (Zuo et al., 2021) | D | 1° | 1 km | GWL | | X | X | X | X | X | | |
| (Girotto et al., 2016) | M | 1° | 36 km | GWS | | X | | X | | X | | OL |
| (Houborg et al., 2012) | M | basin | 4,000 km² | GWS | GWS | X | | X | | X | | OL |
| (Nie et al., 2019) | M | 0.125° | 0.125° | GWS | | X | | X | X | X | | OL |
| (Schumacher et al., 2018) | M | 1,060,000 km² | 0.5° | GWS | | X | | X | X | X | | OL |
| (Tian et al., 2017) | M | 0.5° | 0.5° | GWL | | X | | | | X | | OL |
| (Zaitchik et al., 2008) | M | > 500,000 km² | 4,000 km² | GWS | | X | | X | | X | | OL |

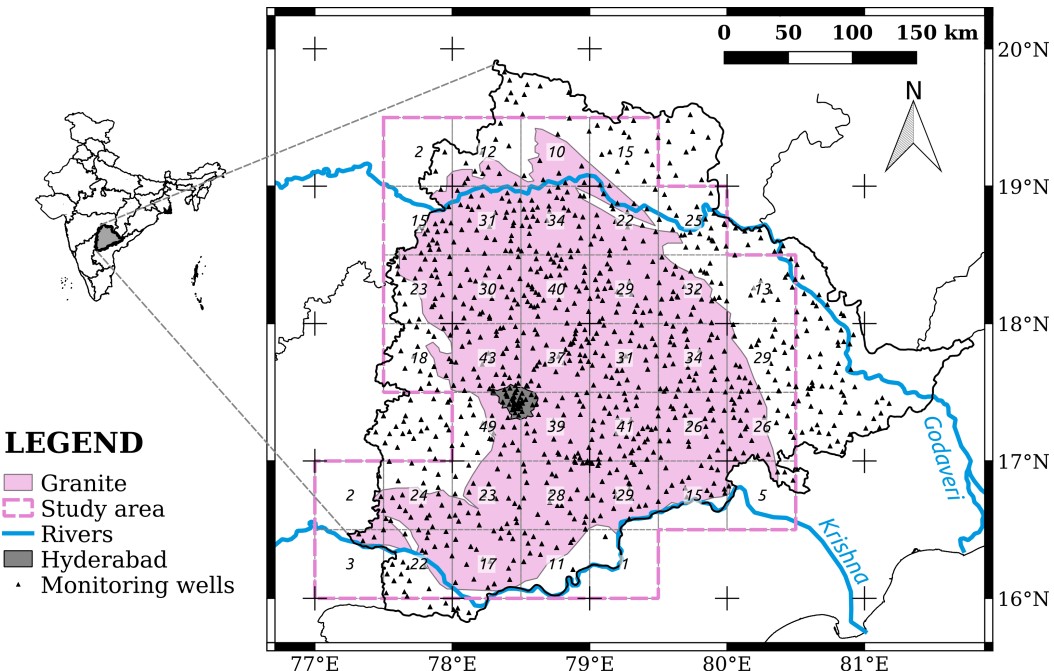

**Figure 1.** Location of the study area (dotted pink line, 113,000 km$^2$) that contours the granitic area of the Telangana State (pink area, 67,000 km$^2$) with the target 0.5° resolution. The number of available observation wells (black triangles) is indicated in the center of each of the 38 0.5° pixels. The grey area indicates the extent of Hyderabad, the capital city of the State.

observation wells in North India (Asoka et al., 2017; Chen et al., 2014; Tiwari et al., 2009), North China (Feng et al., 2013; Huang et al., 2015), Texas (Long et al., 2013) and many other parts of the world (Rodell et al., 2018), it is challenging to
identify a long-term trend for groundwater storage in the Telangana State.

This study focuses on the granitic area of Telangana contoured with the 0.5° resolution GRACE RL06M scaling factor grid. The study area is estimated at 113,000 km$^2$, which is similar to the actual size of a GRACE pixel. Note that the GRACE RL06M pixels were extracted by selecting the 0.5° pixels falling within the granitic area of Telangana (pixels within the pink dotted line in Figure 1).

**2.2 Data**

All data used and their sources are summarized in Table 2. Figure 2 shows time series of some of the data presented below as well as their intra-annual and inter-annual periodicity.

**2.2.1 GRACE TWS**

We used the state of the art GRACE mascon solution from JPL (RL06M) with the Coastal Resolution Improvement (CRI) filter
in this study. The mascon solution uses a priori information derived from near-global geophysical models to prevent striping.



Moreover, it suffers less from leakage errors than the harmonic solution (Watkins et al., 2015) (https://grace.jpl.nasa.gov/data/choosing-a-solution/, accessed on January 19, 2022). Each mascon is $3° \times 3°$, and the scaling factor grid used to restore lost signal has a $0.5°$ resolution. After multiplication of the mascon grid by the scaling factor grid, all the $0.5°$ resolution pixels within the study area are spatially averaged over the study area to produce a LR TWS time series at 113,000 km$^2$ scale. The

baseline of TWS anomalies was modified by subtracting the long-term mean of the 2007-2015 period.

### 2.2.2   Ancillary data

We use ancillary variables from three different datasets to predict GRACE TWS : the monthly rainfall from the TRMM mission, the normalized difference vegetation index (NDVI) from MODIS and the remotely sensed surface soil moisture data from the ESA CCI product (combining passive microwave-derived soil moisture products). All these datasets provide monthly data

except CCI soil moisture (SM CCI) product, which was temporally aggregated at a monthly scale. The temporal window aggregation varies for GRACE TWS and is not always the same than those of ancillary data, but we assumed that the effects of slightly varying windows were negligible. All datasets were aggregated with bilinear resampling both at the downscaling target resolution ($0.5°$) and at regional scale (113,000 km$^2$). The values were converted into anomalies by subtracting the long-term mean of the 2007-2015 period.

### 150   2.2.3   Deconvolution of GRACE TWS with GLEAM

GWS is a sub-compartment of TWS, hence the downscaled GRACE TWS is not directly comparable to in situ-derived GWS. In semi-arid areas, a common assumption is generally made that the essential contributions to TWS are GWS and soil moisture (SM) storage, thus neglecting canopy, snow and surface water storage (Equation (1)):

$$\Delta TWS = \Delta GWS + \Delta SM \tag{1}$$

with $\Delta$ representing the anomalies regarding a baseline, the 2007-2015 average in our case. In the Telangana State, the rivers (except major rivers Godavari and Krishna, see 1) are not perennial and only flow for a few months during and after the monsoon. Surface water stocks are composed of large dams built on major rivers and small reservoirs in upstream part, composing a rainwater harvesting system. The cumulative capacity of the largest dam lakes is estimated to be 113 mm by the Indian National Register of Large Dams (NRLD) (http://cwc.gov.in/sites/default/files/nrld06042019.pdf, accessed on January

19, 2022) and the capacity of the rainwater harvesting system in Telangana was estimated at 30 mm in a previous study (Pascal et al., 2021). This potential reservoir of 143 mm represents 20% of GRACE TWS amplitude in this area during the 2002-2021 time period (710 mm), yet the reservoirs are rarely simultaneously full and, most of the time, surface water storage can be neglected. Most studies use model outputs of SM to deconvolute GRACE TWS, principally GLDAS (Ali et al., 2021; Chen et al., 2019; Jyolsna et al., 2021; Karunakalage et al., 2021; Yin et al., 2018; Zhang et al., 2021b; Zuo et al., 2021) but also

other models like NLDAS (Sahour et al., 2020; Seyoum and Milewski, 2017) or EALCO (Zhong et al., 2021), although (Ning et al., 2014) directly compares downscaled TWS to in situ measurements. We used the Global Land Evaporation Amsterdam Model (GLEAM) v3.5b monthly root zone soil moisture (RZSM) dataset to simulate SM storage which we transform into





**Table 2.** Summary of all data used.

| Variable | Source | Native spatial resolution | Usage |
|----------|--------|---------------------------|-------|
| TWS | GRACE RL06M | 3° | Target variable |
| Rainfall | TRMM 3B43 V7 | 0.25° | Predictor |
| NDVI | MOD13A3v006 | 1 km | Predictor |
| Surface soil moisture | ESA CCI v06.1 passive product | 0.25° | Predictor |
| RZSM | GLEAM v3.5 | 0.25° | Deconvolution of GRACE TWS |
| GWS | Telangana State Groundwater Board | Punctual data | Validation |

anomalies with the baseline 2007-2015. GLEAM v3.5b is a model driven by satellite data that estimates evapotranspiration and soil moisture over a 0.25° resolution grid for the period 2003-2020 (Martens et al., 2017; Miralles et al., 2011). RZSM

anomalies were computed by retrieving the 2007-2015 mean.

### 2.2.4 Validation GWS data (GWS-OW)

We use GWL data from the Telangana State Groundwater Board of India (India Water Resources Information System, https://indiawris.gov.in/wris, accessed on January 19, 2022) for the period 2007-2019. These data provide monthly surveys of instantaneous GWL of 1006 wells distributed over the study area (see Figure 1). Maps of interpolated GWL at 0.5° resolution

were produced and converted in GWL anomaly by retrieving the long-term mean of the 2007-2015 period. These maps were converted to GWS maps by multiplying by a Sy that was calibrated with a linear fitting between GRACE TWS deconvoluted with GLEAM RZSM and the GWL anomaly at regional scale. The Sy was estimated at 4.7%, which is an intermediate and consistent value between the Sy of both layers (saprolite at 10% and fractured granite at 1%) composing the aquifer in the study region. In the following, we designate these so computed GWS anomalies as GWS-OW.

## 3 DOWNSCALING AND VALIDATION METHODS

### 3.1 Evaluation of downscaled data

#### 3.1.1 Gain against the "null hypothesis"

As highlighted in the introduction, a lack in the majority of publications on GRACE downscaling is the comparison of the downscaled GWS with a null hypothesis. In particular, current evaluation methods check that metrics fall in an acceptable

range, but don't check that the downscaled product fits to the validation data better than the LR (original GRACE) product. To fill the gap of current validation strategies of the downscaling methods of GRACE data, new metrics are proposed herein to quantitatively assess the accuracy of the downscaled data compared to the data at the original GRACE resolution (null hypothesis). In this case, two LR TWS references are possible: either the spatially averaged TWS value (produced as explained in Sect. 2.2.1), or the product of the mascon solution and its scaling factor grid at 0.5° resolution. In both cases, the contribution



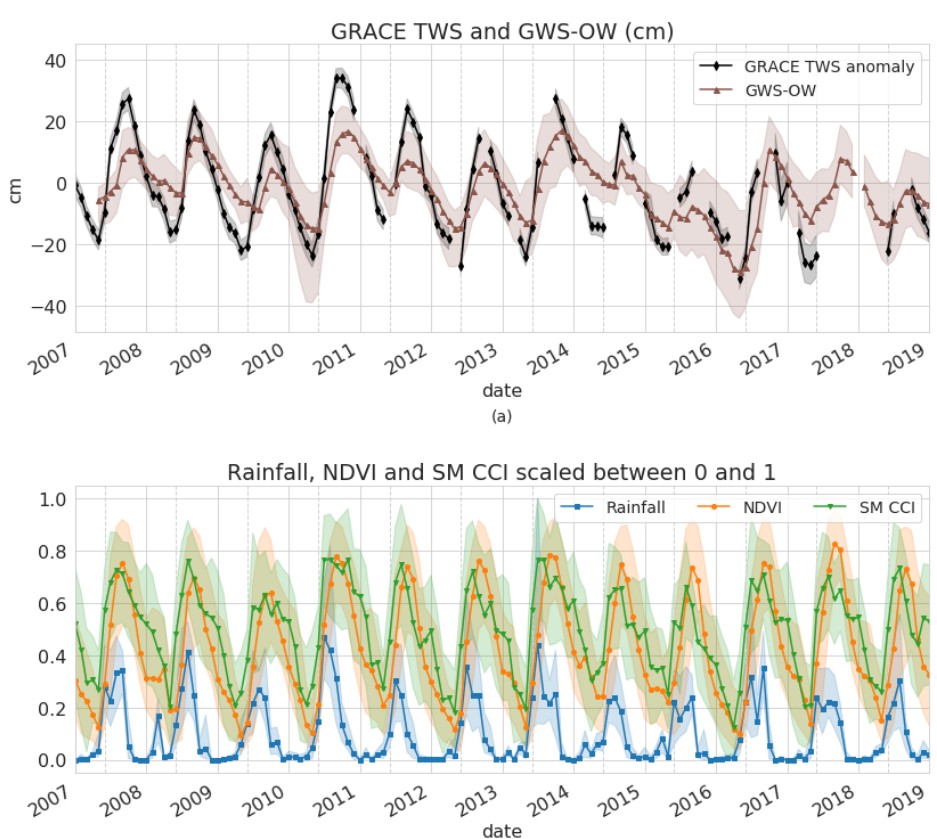

**Figure 2.** Time series of low-resolution (a) GRACE TWS and GWS anomalies derived from in situ measurements (GWS-OW) in cm and (b) rainfall, NDVI and SM CCI. Those three predictors were scaled between 0 and 1 to compare their temporal cycles. The envelope for GRACE TWS is the uncertainty provided with the mascon solution, and those for GWS-OW, rainfall, NDVI and SM CCI correspond to the lowest and highest values found on high-resolution (0.5°) pixels at each time step. The months of June (beginning of the monsoon) are marked by dotted vertical lines.

of SM to TWS is removed (using GLEAM RZSM estimates used at the 0.5° target resolution) to obtain GWS, comparable with in situ data. We chose to use the averaged TWS deconvoluted with the 0.5° GLEAM RZSM (further called GWS-LRref) as the LR reference. The 0.5° scale factor-based product (further called SF) is used as the downscaling first guess whose performance will be compared with the downscaling techniques proposed in this paper.

We chose to compute a relative gain similar to (Merlin et al., 2015). For a given metric M measuring the agreement with the
validation data (e.g. R, RMSE), the gain G is computed as follows (Equation (2)):

$$G = \frac{|M_{opt} - M_{LR}| - |M_{opt} - M_{HR}|}{|M_{opt} - M_{LR}| + |M_{opt} - M_{HR}|} \qquad (2)$$





with $M_{LR}$ the value of the metric for the GWS-LRref, $M_{HR}$ its value for the downscaled GWS, and $M_{opt}$ the optimal value of this metric (e.g. 1 for R, 0 for RMSE). The gain of Equation (2) can be computed in time or in space.

### 3.1.2 Temporal gain at high spatial resolution

For the temporal analysis, we compute this gain on the time series of GWS on all HR pixels and for three metrics : R, $R^2$ and RMSE (Equation (3), (4) and (5)). These are temporal gains, as they measure the improvement of the agreement of the time series on each HR ($0.5°$) pixel where in situ measurements are available.

$$G_{R} = \frac{|1 - R_{LR}| - |1 - R_{HR}|}{|1 - R_{LR}| + |1 - R_{HR}|} \tag{3}$$

$$G_{R^2} = \frac{|1 - R^2_{LR}| - |1 - R^2_{HR}|}{|1 - R^2_{LR}| + |1 - R^2_{HR}|} \tag{4}$$

$$G_{RMSE} = \frac{RMSE_{LR} - RMSE_{HR}}{RMSE_{LR} + RMSE_{HR}} \tag{5}$$

### 3.1.3 Spatial gain at monthly scale

For the spatial analysis, we compare the monthly maps of downscaled GWS with reference maps of GWS-OW. For each time
step, we compute a gain over the LR reference on four metrics : the slope S of the linear regression (Equation (6)), the mean bias B (Equation (7)), R (Equation (3)) and RMSE (Equation (5)).

$$G_{S} = \frac{|1 - S_{LR}| - |1 - S_{HR}|}{|1 - S_{LR}| + |1 - S_{HR}|} \tag{6}$$

$$G_{B} = \frac{|B_{LR}| - |B_{HR}|}{|B_{LR}| + |B_{HR}|} \tag{7}$$

We expect S and R to be closer to 1, and B and RMSE closer to 0 for the downscaled product than for the LR reference. The slope is a common indicator to evaluate downscaled products, in particular for soil moisture downscaling (Merlin et al., 2015; Sabaghy et al., 2020). Indeed, the variability of GWS is expected to be higher at HR and closer to that of in situ measurements than at LR. Computing metrics for each time step rather than on the whole time series (all time steps and all HR pixels mixed) allows for the first time to eliminate the contributions of intra-annual and interannual variations and to specifically isolate the
contribution of GWS spatial variability at the GRACE sampling period.

### 3.2 Statistical downscaling method

We use a data-based downscaling method that consists in training a model at LR between TWS and ancillary variables resampled at LR ($113,000$ km$^2$). This model and ancillary variables at HR ($0.5°$) are then used to predict TWS at $0.5°$. An additive





**Figure 3.** Flowchart of the downscaling method.

correction is applied at LR to force the average of HR TWS predicted by the model to be equal to the TWS observed at LR

(GRACE observation). The corrected TWS at 0.5° is finally deconvoluted into GWS with the GLEAM RZSM. We compare two models often used in the literature: the multilinear regression model and the random forest model. The downscaling process is summarized in the flowchart of Figure 3.

### 3.2.1 Variable selection

For this data-driven approach, we selected remote sensing predictors that have a hydrological meaning. Also we avoided model

outputs, as irrigation is often not well represented in models. The predictors considered herein are: precipitations (TRMM), surface SM (CCI), NDVI (MODIS) as an indicator of crop fraction and the monthly variation of NDVI (ΔNDVI). We also used as predictors the cumulative sum over the past year for all variables (except ΔNDVI), by considering that it provides





information about the state of the aquifer before the start of the irrigation season. Note that some predictors lag behind GRACE
TWS due to the time that hydrological processes take. We determined the optimal time lag between TWS and each variable
from 0 to 3 months by maximizing their temporal correlation coefficients (Sahour et al., 2020; Seyoum and Milewski, 2017).
For both multilinear regression and random forest approaches, parsimonious models are obtained by selecting the optimal
number of the most meaningful variables that allows predicting the TWS. We used the RFECV (recursive feature elimination
with cross-validation) algorithm, which is a greedy feature elimination algorithm similar to sequential backward selection.

### 3.2.2   Multilinear regression model

The multilinear (ML) regression model fits a linear relationship between the target variable $Y$ (here TWS) and $p$ predictors
$X_1, X_2,\ldots X_p$ (Equation (8)):

$$Y = \beta_0 + \beta_1 X_1 + \cdots + \beta_p X_p + E \tag{8}$$

The $\beta_0, \beta_1, \ldots, \beta_p$ are determined by minimizing the mean squared error between the data and the model predictions. This
model has the advantage of being easily interpretable, but is limited by the assumptions that relationships between variables
are linear. Before training the ML model, the issue of multicollinearity (the existence of linear relationships between variables)
was addressed. The elimination of redundant variables increases the precision of the coefficients of the regression, and helps
properly identify the contribution of the remaining variables on the target variable (here TWS), and especially the signs of the
coefficients. We used the Variance Inflation Factor (VIF) (Alin, 2010) as in (Sahour et al., 2020) to detect multicollinearity, and
predictors with a VIF > 10 were removed.

### 3.2.3   Random forest regressor

The random forest (RF) algorithm (Breiman, 2001) is a supervised ensemble learning algorithm composed of independent deci-
sion trees. Each decision tree learns with a subset of the predictors (here the square root of the maximum number of predictors)
using a bootstrap sampling. This method softens the relationship constraints between variables, but loses in interpretability.
There is no need to remove some variables before training the model as the RF algorithm deals well with collinearity.

### 3.2.4   Additive correction

After predicting HR TWS with the ML or RF model, we corrected the TWS values so that the spatial average of HR TWS at
each timestep would be equal to LR TWS. We add an offset value to correct the HR TWS at each month of the time series
that correspond to the difference between LR GRACE TWS and the spatial average of HR TWS predictions at the same date
(Equation (9)):

$$\text{TWS}^{corr}_{HR,t,i} = \text{TWS}_{HR,t,i} + \text{TWS}_{LR,t} - \frac{\sum_i \text{TWS}_{HR,t,i}}{n_{pix}} \tag{9}$$

with $\text{TWS}_{HR,t,i}$ the HR TWS predicted by the model for month $t$ and pixel $i$, $\text{TWS}^{corr}_{HR,t,i}$ the bias-corrected TWS, $\text{TWS}_{LR,t}$
the LR TWS at date $t$, and $\frac{\sum_i \text{TWS}_{HR,t,i}}{n_{pix}}$ the spatial average of HR TWS at date $t$.





**Table 3.** Correlation coefficients of ancillary variables with GRACE TWS. The optimal time lag is indicated by bold correlations. The underlined correlations are not statistically significant.

| Lag | Rainfall | NDVI | SM CCI | $\Delta$ NDVI |
|---|---|---|---|---|
| 0 | **0.90** | 0.30 | 0.79 | 0.10 |
| -1 | 0.85 | 0.68 | **0.91** | 0.51 |
| -2 | 0.54 | **0.79** | 0.71 | **0.68** |
| -3 | 0.16 | 0.68 | 0.34 | 0.63 |

**Table 4.** Variable selection and model performance at low resolution (LR). The number in parenthesis is the number of lag months. Variables with the suffix "cum." are cumulated over the last year. The underlined variables were eliminated with the VIF selection method. The model performance is evaluated against GRACE TWS with the $R^2$ and the RMSE on train and test sets. The $R^2$ with in situ-derived TWS (sum of GWS-OW and GLEAM RZSM at LR) is also shown.

| Model | Variable selection | | | | | | | Model performances | | | |
|---|---|---|---|---|---|---|---|---|---|---|---|
| | Rainfall (2) | NDVI (0) | SM CCI (1) | $\Delta$NDVI (2) | Rainfall cum. | NDVI cum. | SM CCI cum. | RMSE on train set (cm) | RMSE on test set (cm) | $R^2$ with GRACE TWS | $R^2$ with in situ-derived TWS |
| ML | | X | X | X | | | X | 5.2 | 5.0 | 0.90 | 0.78 |
| RF | X | X | X | X | X | | X | 1.9 | 4.6 | 0.97 | 0.82 |

## 4 RESULTS

This section aims at evaluating the efficiency of the two data-based downscaling methods i.e. multilinear (ML) and random
forest (RF) models against GWS-OW. In each case, we compare these results with the first guess downscaling product, i.e. the product of mascon solution and its scaling factors at 0.5° resolution (SF). We analyze the conclusions drawn from the classic evaluation methods found in the literature (Sect. 4.2), then from the new validation method proposed in this study (Sect. 4.3). The synthesis of the different conclusions is presented in Sect. 4.4.

### 4.1 Variables selection and model calibration at LR

The correlation coefficients of the ancillary variables with GRACE TWS at LR are reported in Table 3. The bold correlations indicate which time lag was chosen for each variable: no lag for NDVI, 1 month lag for CCI soil moisture, 2 month lag for $\Delta$NDVI and rainfall. The selected variables for each model are indicated in Table 4. Four variables were selected for the ML model: $\Delta$NDVI, NDVI, SM CCI and SM CCI accumulated over a year. The RF model selected two additional variables: monthly rainfall and rainfall accumulated over a year.


ML and RF models are trained on a random sample of 80% of the whole time series (174 points in total). The selected variables and the model performances are reported in Table 4. The RF model has a better $R^2$ than the ML model (0.97 against 0.90), yet the RMSE on test set is way larger than on train set (4.6 cm against 1.9 cm), revealing that the RF suffers from overfitting. The RMSE on train set is respectively 5.0 and 4.6 cm for the ML model, which represent 7% and 6% of GRACE TWS total amplitude over the region during the study period (71 cm). Both models seem to be able to predict GRACE TWS

with good performance. However, the performance is lower when compared to in situ data. As an example, the $R^2$ between in situ-derived TWS (sum of GWS-OW and RZSM GLEAM) aggregated at LR and GRACE TWS is 0.80, already revealing the uncertainty induced by the deconvolution with GLEAM RZSM. The $R^2$ with in situ-derived TWS falls from 0.90 and 0.97 to 0.78 and 0.82 for the ML and RF predictions respectively. This can be due to the existing uncertainty mentioned earlier, but also to the possible lack of representativeness of in situ measurements at the GRACE spatial resolution.

## 4.2 Classic evaluation


The temporal agreement between GWS-OW and downscaled products was evaluated on every HR pixel with the $R^2$, R and RMSE, for the SF downscaling and both the ML and the RF models with correction by the LR offset value (CORR). Figure 4a shows the spatial distribution of these three metrics for the RF CORR-downscaled GWS for visualization and Figure 4b the distribution of the three metrics on all pixels for all downscaling methods. The temporal agreement of the SF product with

GWS-OW seems to be the worst given the wide distribution of $R^2$ with an average of 0.21 and some outliers in negative values, and an average RMSE of 9.1 cm. The SF product appears to perform less than the LR reference GWS-LRref (average $R^2$, R and RMSE of 0.38, 0.76, 8.2 cm). The R and $R^2$ are better on average for ML CORR (0.79 and 0.42) and RF CORR (0.79 and 0.43), and the reduced variability of R and RMSE for ML CORR and RF CORR suggests that the bias correction produces results with a more uniform quality. The RMSE is still relatively large, ranging from 6.3 to 9.3 cm (resp. 6.4 to 9.3 cm) for the

ML (resp. RF) model. As a reference, the amplitude of GWS-OW in this area during the 2007-2019 time period ranges from 33 to 70 cm on all 38 HR pixels.

## 4.3 Evaluation with temporal and spatial gains

The temporal gains are computed as explained in Sect. 3.1, and are shown for the particular case of RF CORR in Figure 5 for visualization. The spatial gains are computed at each time step between the two points clouds of GWS-OW and GWS-LRref or

downscaled GWS, as illustrated in Figure 6. The boxplots of temporal gains on all HR pixels and the boxplot of spatial gains on the whole time series are shown in Figure 7. ML CORR and RF CORR show the best results: average gains on $R^2$, R and RMSE are respectively 3.2, 6.5 and 1.55% for ML CORR and 4.0, 6.7 and 1.9% for RF CORR. In particular, the temporal gains for the RF CORR product seem to be positive in the North and South of the study area (cf Figure 5), which coincides with the two main rivers basin of the State, but also concerns pixels with mixed geology and where the least amount of observation

wells are available (see Figure 1). The pixel at 17°N, 78°E contains the major part of the capital city of the State Hyderabad, a heavily urbanized area where natural hydrological processes as well as observation wells measurements are highly perturbed by domestic water use, explaining the negative gains on $R^2$, R and RMSE (-4,4, -6.3 and -2.2% respectively).



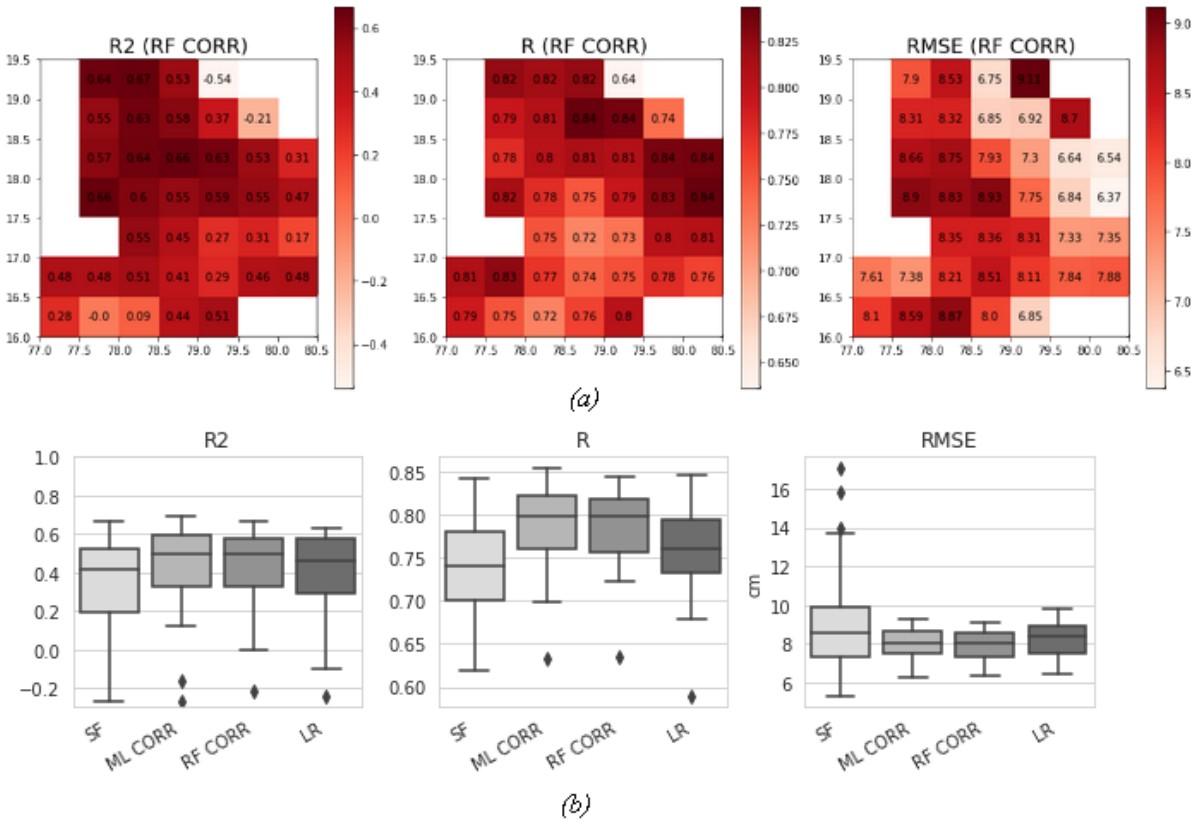

**Figure 4.** (a) Spatial distribution of $R^2$, R and RMSE for the downscaling with the random forest model with bias correction (RF CORR). The numerical value of the metric is indicated in the grid. Abscissa is the East longitude and ordinate the North latitude. (b) Boxplot (median and quartiles) of $R^2$, R and RMSE between GWS-OW and: the scaling factor product (SF), the linear (ML) and random forest (RF) model downscaled products with bias correction (CORR), and the low resolution reference GWS-LRref (LR). The RMSE is an equivalent water height in cm.

In the spatial domain (see Figure 7b), the quality of the SF-downscaled GWS is questionable. The quasi-null gain on bias was expected as the only difference between LR and SF TWS is a multiplicative factor generally close to 1. The SF-downscaled

GWS shows positive gains for slope and R on most of the time series (12.9 and 13.9% respectively on average), but at the cost of a higher uncertainty (-8.5% gain on RMSE). Monthly scatterplots (results not shown) indicate that the slope getting closer to 1 is most of the time a consequence of an increased dispersion due to what appears to be additional noise at each time step brought by the scaling factor grid. The improvement of spatial representativity of GWS with data-based downscaling methods (ML CORR and RF CORR) is shown by majoritarily positive gains on slopes and R (22.9 and 28.8% for ML CORR, 18.2

and 27.2% for RF CORR) while maintaining a general positive gain on RMSE as well (2.0 and 2.2% for ML CORR and RF CORR respectively). The bias correction at LR adjusts the HR TWS predicted by the model to the GRACE TWS amplitude, explaining the null gain on bias for both models.





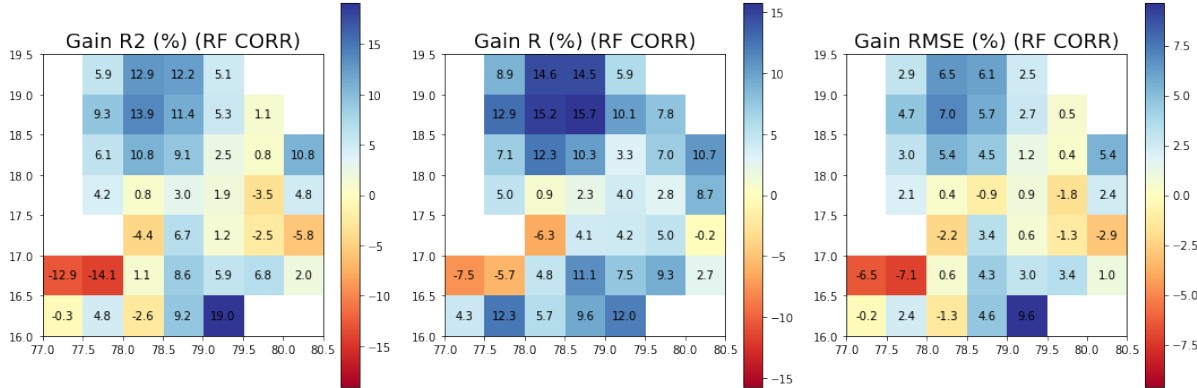

**Figure 5.** Spatial distribution of the gains of $R^2$, R and RMSE for the downscaling with the random forest model with bias correction (RF CORR). The numerical value of the metric is indicated in the grid. Abscissa is the East longitude and ordinate the North latitude.

Complementary to the spatial analysis of temporal metrics (e.g. in Figure 5), the spatial gains can be analysed in time. Figure 8 shows the monthly medians of the gains on slope, R and RMSE for RF CORR downscaled GWS. It appears that gains on slope and R are the lowest during the month of July (beginning of the monsoon). Gains on both slope and R increase until January-

February (beginning of the second crop season that ends in April) and decrease again. The monthly gains of RMSE have lower amplitudes than those of slope and R, but have a similar pattern. We assume that the periodicity in downscaling performances is due to the incapacity of the model trained at LR to restitute the spatial variability of some intermittent processes. During the month of July, heavy rainfall occurs, fills rivers and reservoirs and starts replenishing the empty groundwater stocks. Surface

water is an important component of the water column at this time of year, yet runoff is not directly modeled by any of the variables of the RF model. With the available predictors, it is challenging to restitute the spatial heterogeneity of GWS at HR during this period, given that the spatial variability of SM CCI, and to a lesser extent rainfall, is particularly low in June and July (not shown). The period between February and April, during the dry season, is marked by the heavy pumping and use of surface water for crop irrigation. In April, irrigation stops (the NDVI is at its minimum) and groundwater reach their

lowest level. The model probably fails to restitute the diversity of HR hydrological processes over a heterogeneous zone (e.g. differences in water availability and crop fraction, diversity of crops with different water needs) when water availability and thus water exchanges are very scarce.

## 4.4 Comparative analysis of both validation methods

The thresholds to decide if temporal metrics are poor, satisfactory or good are often arbitrarily decided and are different with

the context of the study and the authors' choices. In our case, all downscaled GWS products have correlation coefficients with GWS-OW systematically larger than 0.57 on each of the 38 HR pixels, which can be considered a quite satisfactory result. With the $R^2$ criteria, ML CORR and RF CORR seem to have the best performance with at least half of the HR pixels having a $R^2$ larger than 0.5. For ML CORR and RF CORR, the RMSE does not go below 6.3 cm and 6.7 cm respectively, with a

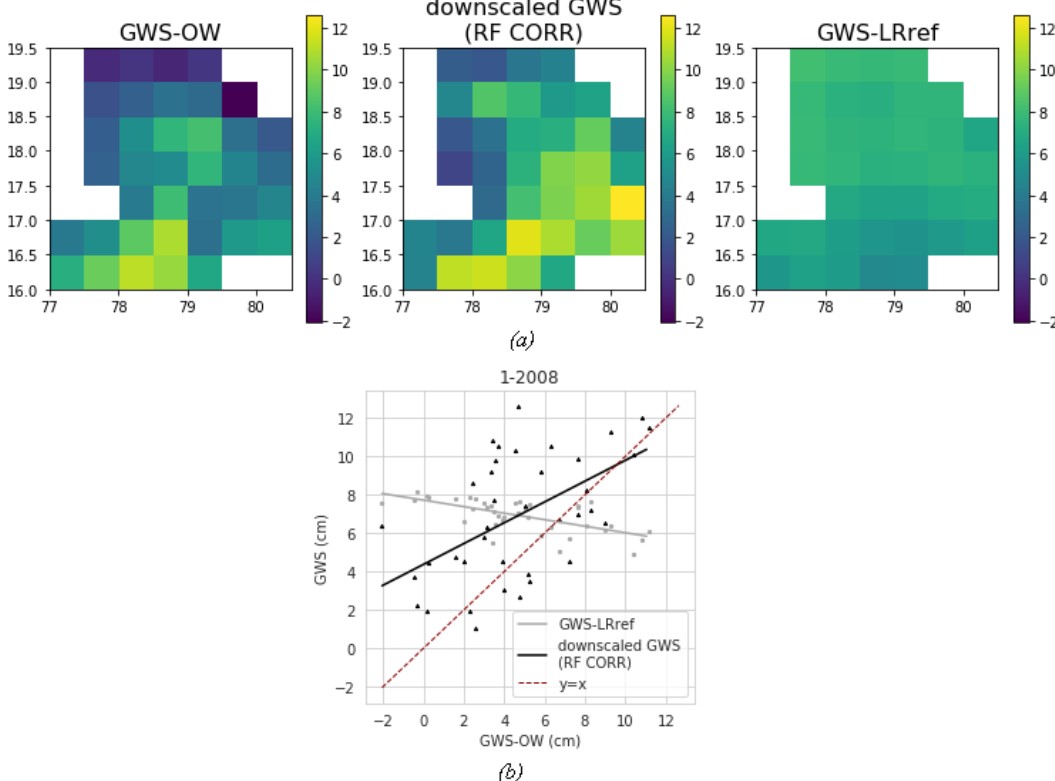

**Figure 6.** Illustration of the spatial gain for the month of January 2008. (a) Maps of HR in situ-derived GWS (GWS-OW), RF CORR-downscaled GWS and LR reference GWS (GWS-LRref). Abscissa is the East longitude and ordinate the North latitude. (b) Scatterplot of the GWS-LRref (grey points) and RF-downscaled GWS (black points) against GWS-OW. The identity function is indicated by a dashed red line. The slope of the two linear regression fits on grey and black points are used to compute the gain on the slope. The difference of dispersion, uncertainty and bias of the two points clouds are evaluated with gains on R, RMSE and mean bias.

median RMSE of 8.1 cm for both methods, which still represent a non-negligible 18% (resp. 16%) error against GWS-OW
(resp. GWS-LRref) amplitude at LR.

Those above appreciations of temporal metrics do not indicate the superiority and the downscaling capacity of these downscaled GWS maps over the GWS-LRref. With spatial and temporal gains, it was shown that ML CORR and RF CORR products are able to improve the temporal agreement with in situ data for most of HR pixels. On the spatial aspect, ML CORR and RF CORR both improve the spatial representativity of GWS for most of the time series (positive gains on slope and R) with
a slightly lower uncertainty (gain on RMSE mostly positive). Additionally to the comparison against the LR reference, the validation in both time and space allows a better understanding of the downscaling strengths and flaws depending on local characteristics of some HR pixels (e.g. presence or rivers, agricultural practices, climatological variability or large cities) or the time of year (e.g. wet or dry season). In the temporal domain, the RF CORR downscaled GWS seems to be better correlated



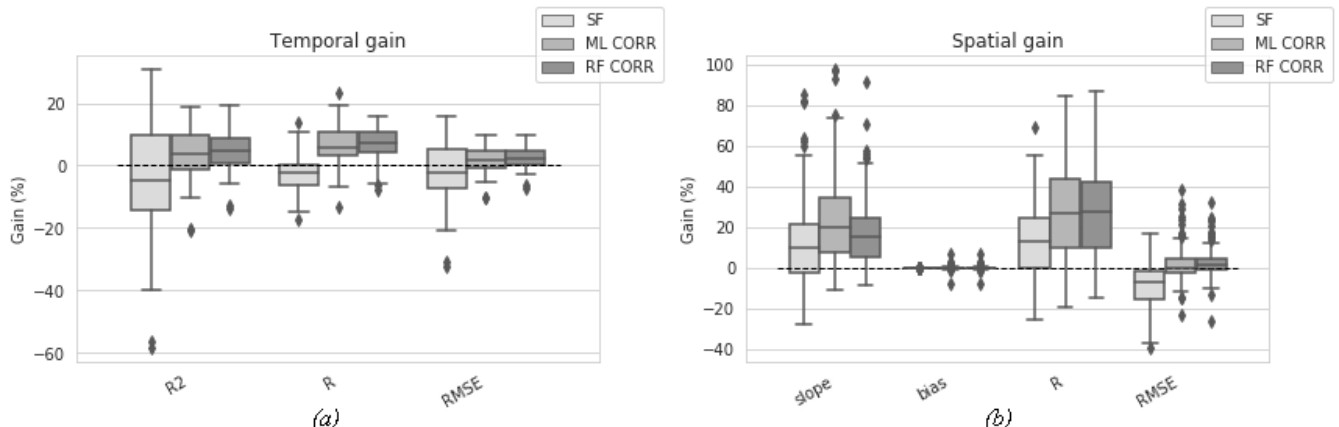

**Figure 7.** Boxplots (median and quartiles) of (a) temporal and (b) spatial gains for the scaling factor (SF), multilinear (ML) and random forest (RF) model downscaling approaches. The designation CORR indicates that the downscaled TWS is corrected for the LR bias from GRACE data.

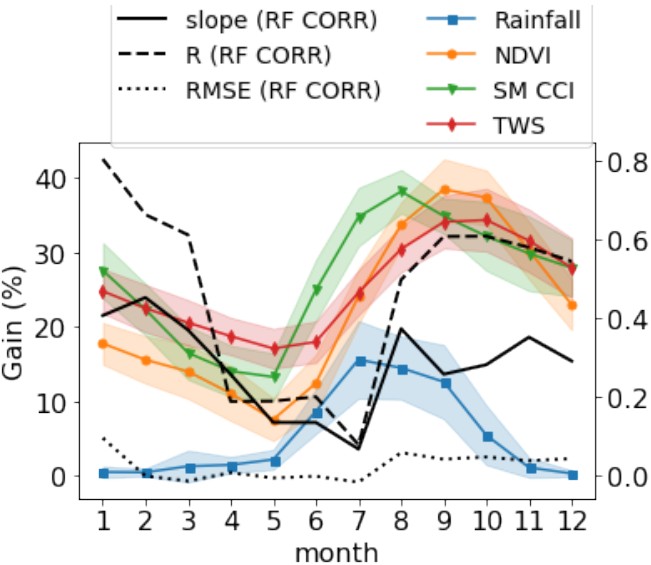

**Figure 8.** Left axis: monthly medians of spatial gains (black curves) on slope, R and RMSE for downscaled GWS with the random forest model with additive correction (RF CORR). Right axis: average ± standard deviation of low-resolution rainfall, NDVI and soil moisture (SM) from the CCI dataset, scaled between 0 and 1.

with in situ data in the North and South of the study area near large rivers. This suggests that the model trained at LR has
difficulties to model certain processes, e.g. the hydrological response to anthropic pressure that is localized and thus smoothed when averaged over a larger region. In the spatial domain, our validation method shows that the RF CORR downscaled GWS



performed less during the dry season and the beginning of the wet season, supposedly because the model fails to represent the spatial variability of GWS when GWS is low or when surface water represents an important fraction of TWS. This highlights the weeknesses of a static model trained on a whole time series with no regards for the specificities of the hydrological

dominant processes at several time of year.

# 5 DISCUSSION

Here we discuss how downscaled products can be impacted by (i) the resolution at which GRACE data are used (Sect. 5.1) and (ii) the uncertainty issues when combining data from heterogeneous sources (Sect. 5.2).

## 5.1 Impact of the GRACE actual resolution on downscaled results

The validation framework proposed in this article was used to evaluate the downscaling potential of the scaling factor built from mascons solution RL06M. The scaling factor is built by fitting a unique factor between the TWS from GLDAS CLM at $0.5°$ and aggregated at mascon scale ($3°$) to evaluate the signal loss over the entire time series. Although it is not meant to downscale the mascon solution, several recent articles in the literature have used the oversampled TWS (spherical harmonics at $1°$ or mascon solution at $0.5°$) as the LR input data. Our validation framework clearly showed that the product of GRACE TWS

mascons and its scaling factor grid (SF method) degrades the temporal agreement with in situ data and is noisy at a monthly time scale. Such results indicate that this product should not be used at the $0.5°$ resolution.

## 5.2 Other uncertainty sources in the validation exercise

Another point that should be highlighted is the difficulty to validate downscaled products with in situ data. Downscaled GWS is built from remote sensing data with various acquisition processes, while validation data are derived from water level depth

acquired by local piezometers with a heterogenous distribution on the study area (see Figure 1). Each methodological steps before a possible comparison between spatialized in situ and remote sensing-derived GWS adds uncertainties at LR, illustrated by low R2 (0.63) and high RMSE (6.1 cm) between GWS-OW and GWS-LRref.

The in situ data have their own uncertainties. First, the GWS derived from GWL measurements is highly dependent on the value of the Sy used. Here we used a horizontally and vertically homogeneous Sy, obtained with a linear adjustment

between LR GWL and GWS-LRref. Some authors avoid this issue by directly comparing GRACE-downscaled GWS with GWL measurements (Karunakalage et al., 2021; Ning et al., 2014; Tian et al., 2019, 2017; Yin et al., 2018; Zhang et al., 2021b, a; Zuo et al., 2021). Another issue is that instantaneous GWL can be impacted by short or long-term effects such as the neighbourhood pumping intensity at the moment of piezometer measurement, which cannot be detected nor rectified with the monthly temporal frequency of acquisition.

The acquisition mode for GRACE is also different from in situ data - unique instantaneous measurement- and other remote sensing predictors - average or sum of higher temporal frequencies products. GRACE has an heterogeneous revisit frequency, where each $300 \times 300$ km$^2$ pixel is informed by approximately 3 overpasses of the GRACE satellites during the month (Tapley





et al., 2004; Zaitchik et al., 2008). This can lead for example to smooth the GRACE anomaly by skipping extremes. Another issue in GRACE acquisition is the exclusively vertical sampling of the gravitational field that produces striping in the solution,

and requires post-processing that alters the signal.

## 6 Conclusions

To date, validation strategies for GRACE-derived downscaled products have rested essentially on the appreciation of temporal metrics or trends between downscaled products and localized in situ measurements. Yet such a validation approach is insufficient to fully assess the usefulness of the downscaling method as it suffers from a lack of (i) appropriate validation of the spatial

distribution of the downscaled GRACE-derived GWS within the GRACE pixel and (ii) comparison with the results that would be obtained without downscaling (by directly using GRACE TWS at the fine scale). This article reviews the validation methods of existing downscaling methods of GRACE data, both model-based and data-based, and proposes a more extensive validation framework. In particular, a set of gains is used to evaluate the improvement of downscaled products against a low-resolution (LR) reference, including both temporal and, for the first time, spatial aspects. Such gains aim at fully determining the quality

and uncertainty of downscaled GRACE-derived GWS products in a more comprehensive way.

The new validation framework is tested to evaluate the performance of two data-based downscaling approaches with multilinear (ML) and random forest (RF) models over a 113,000 km$^2$ fractured aquifer in South India to the target resolution of 0.5°. The HR TWS predicted by each model is bias-corrected at each time step from the difference between the LR GRACE TWS and the average of HR TWS. We use GRACE TWS from the mascon solution RL06M for this study, which is multiplied by its

0.5° scaling factor grid and averaged over the study area to produce the LR reference series. A secondary objective of the paper is to assess as well the downscaling potential of the scaling factor, by considering the product of mascon and scaling factor (SF) at the 0.5° resolution as a downscaled product. The comparison of the two data-based downscaling methods (bias-corrected ML and RF) to the LR reference shows an improvement in terms of correlation with in situ measurements. In the temporal domain, the spatial average gains on Pearson correlation coefficients (R) and root mean squared error (RMSE) is +6.5% and

+1.6% (resp. +6.7% and +1.9%) for ML (resp. RF). In the spatial domain, the gains on R and RMSE is +28.8% and +2% (resp. +27.2% and +2.2%) for ML (resp. RF), respectively. The new validation method also confirms that the SF product cannot be used at 0.5° resolution. Although the average R on HR pixels is similar for all methods (0.74, 0.74 and 0.76 for SF, ML and RF respectively), the SF product degrades both temporal and spatial accuracies at 0.5° resolution compared to the LR (without downscaling) case, showing that it cannot be used as a valid downscaling approach. The spatial analysis of temporal gains

reveals a spatial heterogeneity in downscaling performances, which are particularly poor over urbanized areas. The spatial evaluation originally proposed in this study is also able to analyze the seasonality of the downscaling performance. The RF downscaling performance is lower (gains on R below +10%) during the end of the dry season when GWS is at the lowest, and at the beginning of the monsoon when surface flow, not included in the RF model, is a major process. In particular, the spatial validation presented in this study highlights, for the first time, the flaws of static GRACE dowscaling methods in contexts

where the dominant hydrological processes are not the same throughout the year (such as a highly irrigated semi-arid region



with a wet and a dry season as in the case of this study). This shows how complete and comprehensive validation approaches are an essential tool to interpret spatially and temporally the quality and uncertainty of the downscaled GRACE-derived GWS products, and hence to better understand and improve downscaling models and their hypotheses in the future.

While GRACE-FO mission provides continuity of spaceborne gravity change measurements, upcoming similar missions 420 (MARVEL (Lemoine and Mandea, 2020; Lemoine et al., 2020), MAGIC (Massotti et al., 2021)) plan to significantly improve the precision and quality of gravimetric estimates by proposing new configurations for the satellite constellations. Neverthe-less the specified spatial resolution for those future data still undergo strong technical limitations. Therefore, the recourse to downscaling techniques will be, at least in the medium term, the only way to obtain TWS products at a finer scale useful for basin-scale water management.

*Author contributions.* The conceptualization of this work and the methodology was developed by Claire Pascal and Olivier Merlin. Experiments and analysis were made by Claire Pascal and supervised by Olivier Merlin and Sylvain Ferrant. The manuscript was written by Claire Pascal with contributions from Olivier Merlin, Sylvain Ferrant, Adrien Selles and Jean-Christophe Maréchal. Water level data were provided by Abhilash Paswan and cleansed by Adrien Selles.

*Competing interests.* No competing interests are present.

*Acknowledgements.* This research was realized as part of a PhD thesis funded by a French national research ministry doctoral fellowship.





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
