# Peer review of "Evaluating downscaling methods of GRACE data: a case study over a fractured crystalline aquifer in South India"

_Hydrology and Earth System Sciences, 2022_

## Author Comment (AC1)

**Response to Anonymous Referee #1**

**Authors:** The authors warmly thank the reviewer for their careful review of the paper and positive comments on the proposed study. These comments are all valuable and helpful for revising and improving our paper, and we have studied them carefully. We respond to the reviewer's comments below and we detail how we plan on improving the paper.

**General comment**

The paper presents a method for a non-static temporal and spatial validation of the downscaled GRACE (Gravity Recovery and Climate Experiment) data at a resolution deemed appropriate for assessing groundwater storage and irrigation. The authors have combined in situ measurements (e.g groundwater level (GWL) with data-driven (e.g Random Forest and ML) models within an extensive validation framework. Their motivation was driven by the lack of comprehensive dynamic validation strategies for GRACE-derived downscaled products in both time and space to cope with changing hydrological processes through the seasons. In general, their results show that the bias-corrected ML and RF improved the correlation with in situ measurement as compared to the to the LR reference. However, the scaling factor method (SF) degraded the performances and cannot be used at 0.5° resolution as a valid downscaling approach. They also highlighted the flaws of static GRACE dowscaling methods in catchments or areas with hydrological processes varying across the year.

I found the paper very nice to read and has some novelty in both philosophy and methodology dealing with downscaling the GRCAE product which are of interest to the audience of HESS. It is well written and structured in coherent sections with appropriate content.

- By start reading the paper the sentence L 6-8 in the abstract " The point is that the performance of GWS downscaling methods may vary in time due to changes in the dominant hydrological processes through the seasons. To fill the gap, this study investigates the dynamic performance of GWS downscaling by developing a new metric for estimating the downscaling gain (new validation) against non-downscaled GWS" draw my attention. This is one of the main motivations behind this work. However, I was not able to see an explicit consideration of the variability of the dominant hydrological processes in the proposed methodology nor in the results and discussions. If this has been done within the GLEAM model to simulate the soil moisture (SM) storage then this deserves better description and thorough discussion in the results and discussion section.

**Authors:** We acknowledge that this issue was unclear in the previous version. To address this concern, more detailed and specific discussions will be inserted in section 4.3 of the revised manuscript. In particular, we better describe the dominant hydrological processes throughout the year and how they are (or are not) represented by the model. We also modify the figure 8, by adding two new complementary graph (8b and 8c in the revised).

New paragraphs (section 4.3 of the revised):

[revised manuscript text omitted]

Detailed comments

- It may be better to add the native resolution of the SM CCI product to the text (L. 145). Sure, the information exists in the Table 2.

**Authors:** This information (0.25°) will be added to the main text of the revised version (section 2.2.2).

- Why there is a need to check whether the downscaled product fits to the validation data better than the LR (original GRACE) product?

**Authors:** Thanks to the reviewer for highlighting this point. Comparing the fit to the validation data of both downscaled GWS and LR GWS allows to assess whether the downscaling process is necessary at all. Indeed, there is no need to use a downscaled product over the LR product if it gives poorer performances at the fine resolution when compared to validation data. What we regret in state-of-the-art validation methods is that performance metrics between downscaled and validation data are qualitatively considered satisfying or unsatisfying. Using a "reference hypothesis" (here "non-downscaled" case) allows to have a reference and to quantitatively judge whether the downscaled GWS is better or worse in terms of accuracy at the targeted (fine) resolution. This will be better explained in the manuscript in Section 3.1.1:

*"As highlighted in the introduction, a lack in the majority of publications on GRACE downscaling is the comparison of the downscaled GWS with a null hypothesis. In particular, current evaluation methods check whether metrics fall within an acceptable range that is qualitatively defined. Using a "reference hypothesis" (here the "non-downscaled" case) allows to quantitatively judge whether the downscaled GWS is better or worse in terms of accuracy at the targeted (fine) resolution, and to evaluate if the downscaling process is efficient."*

- In Fig. 2. How the uncertainty envelope was calculated? Can you add this to the text in the appropriate section?

**Authors:** The uncertainty envelope is the average of the mascon uncertainty resampled at 0.5° that is provided with GRACE data. This information will be added in the legend of Figure 2a in the revised manuscript.

- In L. 276 you reported that " …revealing that the RF suffers from overfitting". Firstly, can you add the R2 value for the test set in the RF? Secondly, do you think the data quality is responsible for the overfitting of the RF during the test phase?

**Authors:** The R2 for test and train sets for LM and RF models are :
RF : R2 test = 0.93 ; R2 train = 0.98
LM : R2 test = 0.91 ; R2 train = 0.89
We agree with the reviewer that overfitting can be due to data quality, as the model learns data noise, but overfitting also partly comes from the small amount of data available. In fact, the model is trained on an ensemble of 139 points, which is relatively small compared to the complexity of the RF model, resulting in poor generalization.
The R2 values obtained for train and test sets will be added to the revised manuscript in section 4.1. and in table 4:

"*The RF model has a better R2 than the ML model (0.97 against 0.90), yet the RMSE on the test set is way larger than on the train set (4.6 cm against 1.9 cm).* This reveals that the RF model suffers from overfitting due to data quality and the small amount of data (139 points) used to train the model, resulting in poor generalization."

L. 280 "…already revealing the uncertainty induced by the deconvolution with GLEAM RZSM". I don't understand how the lower performance as compared to in situ is attributed to the uncertainty? This needs to be clarified. In addition, I think that there is a need for better developing the uncertainty issue in this paragraph. This deserves better discussion here although a section on other uncertainty sources in validation already exists in the discussion.
**Authors:** We thank the reviewer for raising this issue. What we meant is that we cannot expect a high performance when comparing satellite data (or modeled from satellite data) to in situ data, because of (i) the inherent uncertainties of the data, (ii) the interpolation of in situ data and more generally (iii) the diversity of data sources. All those uncertainty sources also apply to the TWS predicted by models at both low and high resolutions. For clarity, the sentence at Line 280 will replaced in the revised by:

"However, the performance is lower when compared to in situ data. As an example, the R2 between in situ-derived TWS (sum of GWS-OW and RZSM GLEAM) aggregated at LR and GRACE TWS is 0.80. This shows that only limited agreement can be expected between satellite data (or modeled from satellite data) to in situ data, because of (i) the inherent uncertainties of the data, (ii) the interpolation of in situ data and more generally (iii) the diversity of data sources. All those uncertainty sources also apply to the TWS predicted by models at both low and high resolutions."

---

## Author Comment (AC2)

**Response to Anonymous Referee #2**

**Authors:** The authors warmly thank the reviewer for their careful review of the paper and positive comments on the proposed study. These comments are all valuable and helpful for revising and improving our paper, and we have studied them carefully. We respond to the reviewer's comments below and we detail how we plan on improving the paper.

The authors present a new approach to interpret the spatial and temporal quality of uncertainty in downscaled GRACE products. The paper is well stuctured and methods are clearly presented.

I have only a few minor comments:

- English needs to be checked

**Authors:** An effort was made to carefully check the English and the revised version will be corrected accordingly.

- Inline citation style needs to be corrected

**Authors:** We thank the reviewer for pointing this out. Citation style was changed on line 165 (Ning et al., 2014), line 194 (Merlin et al., 2015), and line 248 (Sahour et al., 2020).

- Please add data sources to Figure 1

**Authors:** The geological map was obtained from Phani (2014), and the observation wells coordinates from India Water Resources Information System (WRIS, https://indiawris.gov.in/wris, last accessed on January 19th 2022). This information will be added to the legend of Figure 1.

- It is not clear how the well data have been interpolated and how this affects the analysis

**Authors:** We thank the reviewer for highlighting this. Well data were interpolated with the Inverse Distance Weighting (IDW) method. IDW avoids kriging bias, which may come from a lack of representativeness of the well data, the incertainty in topographic data, etc. IDW is also more accurate than kriging on data points as it suffers less from modeling uncertainties. However, the interpolation method does not have a significant impact at the 0.5° resolution. This will be added to the new version of the manuscript (section 2.2.4):

"Maps of GWL at 0.5° were produced from the interpolation of well data with the Inverse Distance Weighting (IDW) method (which avoids kriging bias and provides more accurate values on data points), and were converted in GWL anomaly by retrieving the long-term mean of the 2007-2015 period."

**Bibliography**

Phani, R.C., 2014. Mineral Resources of Telangana State, India: The Way Forward. Int. J.
Innov. Res. Sci. Eng. Technol. 3, 15450–15459.
https://doi.org/10.15680/IJIRSET.2014.0308052

---

## Author Response (AR1)

**Comments to the author**:

Reviewers provided positive comments on the paper. Authors made changes on the manuscirpt according to the comments. Authors are required to elaborate further on the need and new contribution for this study, and have a thorough editing on the conciseness and cohesiveness between sections of the paper.

**Author's response**

An introductory paragraph to section 3 was added to improve the cohesiveness between sections. An effort was made to shorten the paper, in particular two paragraphs were deleted (lines 52-59 and 162-166 of the previous version of the manuscript). Several paragraphs were added on section 4.3 following reviewer 1's remarks, however we believe that they are essential for the analysis and shows more clearly the contribution of this study (ability of the validation method to analyse the performance of the downscaling process in terms of dominant hydrological processes and their seasonal variability). More explanation were given in lines 79-81 to explain how comparing performance metrics to a low resolution reference is crucial in evaluation GRACE downscaling methods.